# Single-cell whole-genome sequencing, haplotype analysis in prenatal diagnosis of monogenic diseases

Liang Chang[1,2,3,4,*], Haining Jiao[5,*], Jiucheng Chen[6], Guanlin Wu[6], Ping Liu[1,2,3,4], Rong Li[1,2,3,4], Jianying Guo[1,2,3,4], Wenqing Long[5], Xiaojian Tang[5], Bingjie Lu[6], Haibin Xu[6], Han Wu[6]

**Monogenic inherited diseases are common causes of congenital disabilities, leading to severe economic and mental burdens on affected families. In our previous study, we demonstrated the validity of cell-based noninvasive prenatal testing (cbNIPT) in prenatal diagnosis by single-cell targeted sequencing. The present research further explored the feasibility of single-cell whole-genome sequencing (WGS) and haplotype analysis of various monogenic diseases with cbNIPT. Four families were recruited: one with inherited deafness, one with hemophilia, one with large vestibular aqueduct syndrome (LVAS), and one with no disease. Circulating trophoblast cells (cTBs) were obtained from maternal blood and analyzed by single-cell 15X WGS. Haplotype analysis showed that CFC178 (deafness family), CFC616 (hemophilia family), and CFC111 (LVAS family) inherited haplotypes from paternal and/or maternal pathogenic loci. Amniotic fluid or fetal villi samples from the deafness and hemophilia families confirmed these results. WGS performed better than targeted sequencing in genome coverage, allele dropout (ADO), and false-positive (FP) ratios. Our findings suggest that cbNIPT by WGS and haplotype analysis have great potential for use in prenatally diagnosing various monogenic diseases.**

## Introduction

Monogenic disorders typically result from a single gene lesion. Although most individual monogenic diseases are rare, combined, they affect 10 in 1,000 births (WHO | human genomics global health, 2019), representing a substantial threat to human health. As of 12 July, 2022, the Online Mendelian Inheritance in Man (OMIM) database (https://www.omim.org/statistics/geneMap) has reported 6,134 monogenic diseases with known pathogenetic mutations, involving 4,288 genes. Therefore, it is necessary to monitor these genetic risks through prenatal examinations to better manage potential congenital abnormalities caused by genetic diseases. The first prenatal diagnostic tests for genetic diseases were developed 60 yr ago. Chorionic villus sampling and amniocentesis coupled with advanced cytogenetics and molecular diagnosis can now detect most diseases with known genetic causes (Steele & Breg, 1966; Nadler & Gerbie, 1970; Simoni et al, 1983; Levy & Stosic, 2019). Unfortunately, these prenatal tests are invasive and pose a potential risk to the fetus; gestational loss occurs in <1% of individuals undergoing such tests (Simpson, 2012; Alfirevic et al, 2017; Salomon et al, 2019). Moreover, in some underdeveloped countries and regions, it is difficult to perform invasive examinations regularly because of the lack of medical resources.

Noninvasive prenatal testing (NIPT), an important risk-free prenatal examination with improving accuracy, is becoming increasingly popular in clinical practice (Brady et al, 2016). Remarkable achievements have recently been made in detecting prenatal diseases with NIPT based on cell-free fetal DNA (cffDNA) isolated from maternal peripheral blood (Rabinowitz & Shomron, 2020). However, various risk factors in cffDNA-based NIPT, including a limited detection rate for chromosome structural abnormalities and single gene mutations, the low concentration and instability of cffDNA in maternal blood, fetal/placental mosaicism, and maternal chromosome abnormalities, can lead to inaccurate test results (Spits & Sermon, 2009; Beaudet, 2016; Rabinowitz & Shomron, 2020). In contrast to cffDNA, the rare circulating fetal cells in the maternal blood, mainly circulating trophoblast cells (cTBs) and fetal nucleated red blood cells (fNRBC), represent unique sources of fetal DNA without maternal interference (Zipursky et al, 1959; Walknowska et al, 1969; Herzenberg et al, 1979; Beaudet, 2016; Singh et al, 2017). Given the recent progress in single-cell genomics, researchers have explored cell-based NIPT (cbNIPT) because of the advantage of studying pure and intact fetal genetic material from fetal cells. It has been demonstrated that fetal cells captured from maternal blood can be used for subsequent sequencing to detect chromosomal (Breman et al, 2016) and subchromosomal (Kolvraa

[1]Center for Reproductive Medicine, Department of Obstetrics and Gynecology, Peking University Third Hospital, Beijing, China  [2]National Clinical Research Center for Obstetrics and Gynecology (Peking University Third Hospital), Beijing, China  [3]Key Laboratory of Assisted Reproduction (Peking University), Ministry of Education, Beijing, China  [4]Beijing Key Laboratory of Reproductive Endocrinology and Assisted Reproductive Technology, Beijing, China  [5]Department of Obstetrics and Gynecology, Rui-Jin Hospital, Shanghai Jiao Tong University School of Medicine, Shanghai, China  [6]Unimed Biotech (Shanghai) Co., Ltd., Shanghai, China

Correspondence: changliangchina@163.com; h.wu@unimeddx.com
*Liang Chang and Haining Jiao are first authors

et al, 2016; Hou et al, 2017; Vossaert et al, 2019) abnormalities after DNA extraction and amplification.

Our previous study confirmed that the targeted sequencing of a 67-gene panel combined with a haplotype analysis could detect monogenic diseases (e.g., congenital deafness and ichthyosis) from individual cTBs captured from the maternal peripheral blood (Chang et al, 2021). Whole-genome sequencing (WGS) has certain advantages over targeted NGS for detecting mutations beyond the targeted regions and complex structural variations (Palmer et al, 2021). However, few studies on WGS applications at the single-cell level and performance surveys comparing WGS with other sequencing methods have been conducted in clinical settings. Because of the limited genetic material, single-cell DNA sequencing usually requires whole-genome amplification (WGA), leading to biases in the sequencing data, such as non-uniformity of genome coverage and high allele dropout rates (Zhang et al, 2015; Volozonoka et al, 2022). These intrinsic limitations complicate downstream analyses, including genomic variant detection (Satas & Raphael, 2018), and prevent direct mutation analysis. Haplotype analysis could be beneficial in this scenario (Clark, 1990). Common haplotype analyses mainly use population data or multiple cells for phasing (Kumar et al, 2015; Guo et al, 2018), but when the number of cells is small, and there are no population data, only relative phasing can be conducted using DNA from the father, mother, and proband. Whether single-cell–based WGS combined with haplotype analysis can be used for clinical diagnosis deserves further investigation.

In this study, we verified the feasibility and scope of single cTB WGS combined with haplotype analysis for examining monogenic diseases inherited from the parents using cTBs and samples from affected family members including the proband, providing the first insight into the potential application of WGS-based cbNIPT in prenatal diagnostics.

# Results

## Clinical information of recruited families

A total of four families (one family with deafness, one with hemophilia, one with large vestibular aqueduct syndrome [LVAS], and one healthy control) were included in this study. Except in the deafness family, described in the study by Chang et al (2021), all of the women conceived naturally. In the hemophilia family (Fig S1A), the proband (child) carries a hemizygous mutation of F9 gene c.424G>T on ChrX, which was inherited from the mother. In the LVAS family (Fig S1B), the proband (child) carries compound heterozygous mutations of SLC26A4 gene c.1975G>C and c.281C>T on Chr7; c.281C>T was inherited from the father and c.1975G>C from the mother. Details of all the included families are listed in Table 1.

## Capture and confirmation of cTBs

In all cases, the peripheral blood of pregnant women was subjected to procedures to capture cTBs (see the Materials and Methods section). The deafness family had compound heterozygous mutations in the Chr13:GJB2 gene (NM_004004.5; c.235delC [p.Leu79Cysfs] and c.299_300delAT [p.His100Argfs]); the details regarding cTB

isolation (CFC178) and STR identification in this family are described in a previous study (Chang et al, 2021). After manual confirmation of candidate cells, one or two top candidate cTBs were chosen from the hemophilia, LVAS, and healthy families (Fig S2A). Cell CFC616 was obtained from the family with hemophilia. Cell CFC111 was obtained from the family with LVAS. Cells CFC518 and CFC2282 were isolated from the peripheral blood of pregnant woman from the healthy family. A white blood cell was stained as the control. The candidate trophoblast cells were successfully obtained for the subsequent single-cell analysis.

After single-cell WGA, STR analysis was performed to confirm the origin of candidate cTBs. Representative paternal-specific alleles are shown in Fig S2B, and the data indicated that CFC616, CFC111, CFC518, and CFC2282 were cTBs. In CFC616 from the peripheral blood of a patient carrying hemophilia, 75% (12 out of 16) STR loci were successfully detected and five paternal-specific alleles were identified (Table S1). In the CFC111 cell from the LVAS family, the detection rate of STR loci was 43.75% (7 out of 16), and three paternal-specific alleles were identified (Table S2), confirming the cell's fetal origin. CFC518 and CFC2282 were isolated from the healthy family, and the detection rates of STR loci were 87.50% (14 out of 16) and 81.25% (13 out of 16), respectively; six and five paternal-specific alleles were identified, respectively (Table S3). Not all STR loci can be identified from single-cell WGA products, likely because of allele dropout (ADO) or PCR failure. Overall, we successfully isolated cTBs from all families with monogenic diseases and the healthy family.

## Sequencing depth test in the healthy family

WGS data from healthy family's genomic DNAs (gDNAs) and single cells were randomly subsampled at various depths to test the sequencing depth required for appropriate downstream analysis (see the Materials and Methods section). In the WGS data from gDNAs, the genome coverage remained stable at over 90% (Fig 1A and B and Table S4). In the captured cTBs CFC518 and CFC2282, the genome coverage of their single-cell WGS data increased rapidly until the sequencing depth exceeded 15X (Fig 1C and D and Table S4). In addition, when the sequencing depth was lower than 15X, the false-positive (FP) ratios and ADO of CFC518 and CFC2282 decreased with the increase in the sequencing depth (Fig 1E–H and Table S4). Finally, the genome coverage and the number of covered genes in the targeted regions (67-gene panel, whole-exome region, and OMIM gene panel region) by WGS suggested that the coverage of CFC518 and CFC2282 increased linearly with the sequencing depth up to 15–20X (Fig 1I–T and Table S4). These results suggest that a ~15X sequencing depth is adequate for WGS of gDNA and single-cell WGA products in terms of genome coverage, FDR, and ADO. Therefore, in the subsequent analysis of families with monogenic disease, WGS of individual samples was conducted at a sequencing depth of ~15X.

## Single-cell haplotype phasing

The deafness and hemophilia families, unlike the LVAS family, had paired amniotic or fetal villi samples available as references; thus, we could use the ADO and FPR to evaluate the sequencing quality. The FPR (deafness: 30.2%, hemophilia: 26.9%) and ADO (deafness: 17.3%, hemophilia: 18.5%) indicated that the sequencing quality of

**Table 1. Clinical and molecular diagnostic information of the four recruited families.**

| Family | Member | Gender | Age (yr) | Height (cm) | Body weight (kg) | Fertilization method | Pregnancy history | Sample | Type | Genes | Pathogenic loci |
|---|---|---|---|---|---|---|---|---|---|---|---|
| Deafness | Mother | Female | 36 | 156 | 66 | In vitro fertilization | Previous pregnancy: 3; spontaneous abortion: 0; full term birth: 0; number of children: 1; artificial abortion: 1 | Blood | gDNA | *GJB2* | Chr13: c.299_300delAT, heterozygote |
| | | | | | | | | cTBs-CFC178 | WGA | *GJB2* | Chr13:c.235delC, heterozygote |
| | | | | | | | | Amniotic | gDNA | *GJB2* | Chr13:c.235delC, heterozygote |
| | Father | Male | 41 | 171 | 76 | | | Blood | gDNA | *GJB2* | Chr13:c.235delC, heterozygote |
| | Proband | Male | 11 | 154 | 42 | | | Blood | gDNA | *GJB2* | Chr13: c.299_300delAT, heterozygote |
| Hemophilia | Mother | Female | 34 | 162 | 60 | Natural | Previous pregnancy: 1; spontaneous abortion: 0; full term birth: 1; number of children: 1; artificial abortion: 0 | Blood | gDNA | *F9* | ChrX:c.424G>T, heterozygote |
| | | | | | | | | cTBs-CFC616 | WGA | *F9* | ChrX:c.424G>T, hemizygote |
| | | | | | | | | Fetal villi | gDNA | *F9* | ChrX:c.424G>T, hemizygote |
| | Father | Male | 39 | 165 | 72 | | | Blood | gDNA | *F9* | Wildtype |
| | Proband | Male | 5 | 112 | 18 | | | Blood | gDNA | *F9* | ChrX:c.424G>T, hemizygote |
| LVAS | Mother | Female | 28 | Unknown | Unknown | Natural | Previous pregnancy: 4; spontaneous abortion: 0; full term birth: 1; number of children: 1; artificial abortion: 2 | Blood | gDNA | *SLC26A4* | Chr7:c.1975G>C, heterozygote |
| | | | | | | | | cTBs-CFC111 | WGA | *SLC26A4* | Chr7:c.281C>T, heterozygote |
| | Father | Male | 27 | Unknown | Unknown | | | Blood | gDNA | *SLC26A4* | Chr7:c.281C>T, heterozygote |
| | Proband | Male | 5 | Unknown | Unknown | | | Blood | gDNA | *SLC26A4* | Chr7:c.281C>T, heterozygote Chr7:c.1975G>C, heterozygote |
| Health | Mother | Female | 29 | Unknown | Unknown | natural | Previous pregnancy: 0; spontaneous abortion: 0; full term birth: 0; number of children: 0; artificial abortion: 0 | Blood | gDNA | | |
| | | | | | | | | cTBs-CFC518 | WGA | | |
| | | | | | | | | cTBs-CFC2282 | WGA | | |
| | Father | Male | Unknown | Unknown | Unknown | | | Blood | gDNA | | |
| | Fetus | Unknown | Unknown | Unknown | Unknown | | | Saliva | gDNA | | |

single cells was compromised because of technical limitations; thus, direct SNP detection may not reliably determine whether single cells carry pathogenic mutations. Therefore, haplotype analysis was required for further analysis. In the deafness family, CFC178 carried the pathogenic haploid P1 (10 upstream key SNPs, three downstream key SNPs) from the father and the nonpathogenic haploid M2 (five upstream key SNPs, five downstream key SNPs) from the mother (Fig 2A and Table S5), consistent with the results from the paired amniotic fluid. Because the causative gene F9 in the hemophilia family was inherited on the X chromosome and the sex of the CFC616 single cell was male, we only needed to consider whether the cell carried the causative gene from the mother. The results (Fig 2B and Table S6) suggested that CFC616 carried the haploid M1 (six upstream key SNPs, seven downstream

key SNPs) of the maternal pathogenic loci, consistent with the fetal villi sample. The haplotype analysis of the LVAS family (Fig 2C and Table S7) revealed that CFC111 carried both the paternal and maternal pathogenic chromatid P1 (nine upstream key SNPs, 26 downstream key SNPs) and M1 (23 upstream key SNPs, seven downstream key SNPs). These data suggest that ~15X WGS per single cell is sufficient for downstream haplotype analysis to accurately predict the heredity of monogenic diseases.

## Comparison of WGS and targeted sequencing

For each single cell from the three families with monogenic diseases (CFC178 from the deafness family, CFC616 from the hemophilia family, and CFC111 from the LVAS family), ~200X targeted

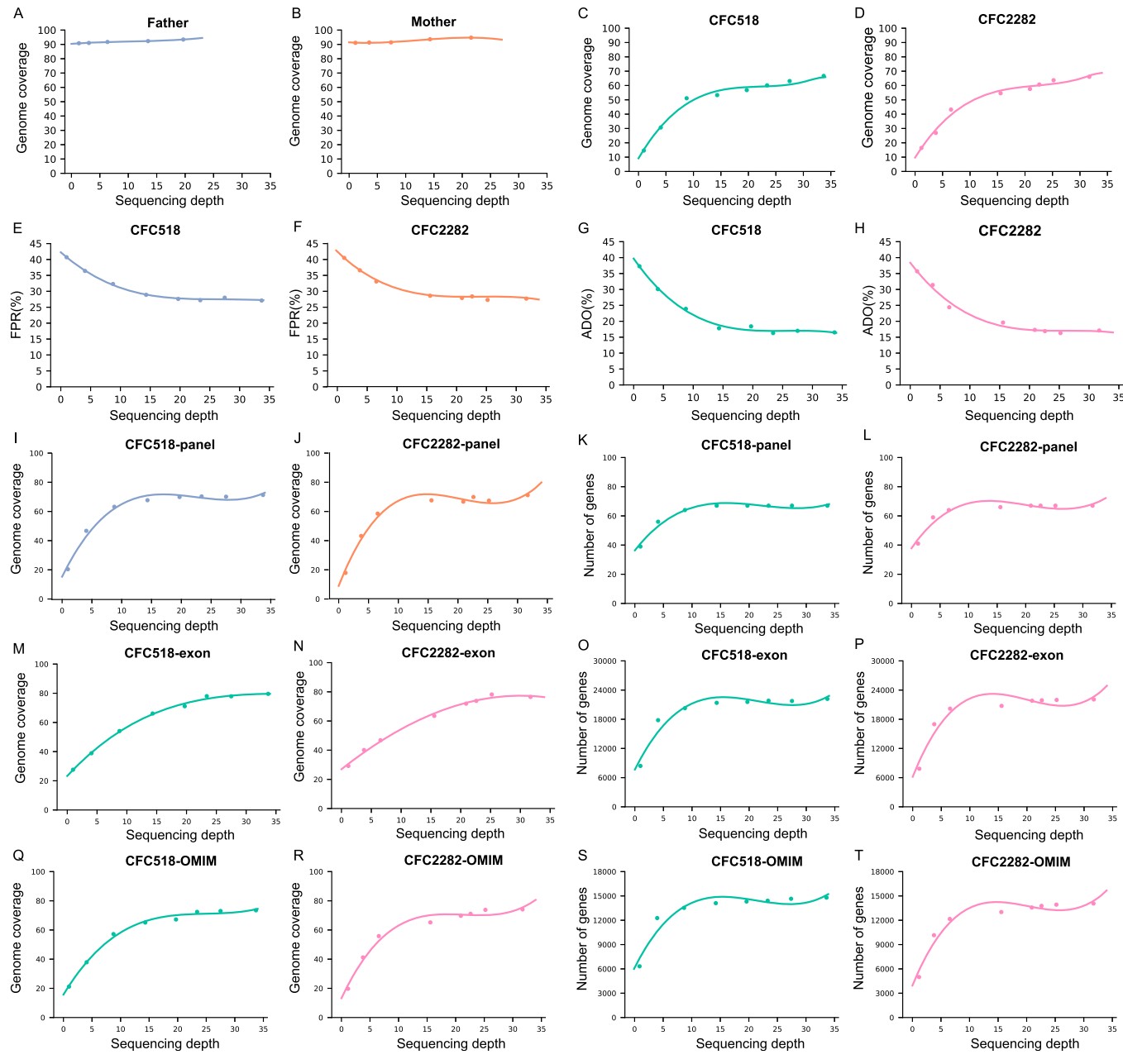

**Figure 1.  Sequencing depth test in the healthy family.**
**(A, B)** Genome coverage of different sequencing depths for the father (A) and the mother (B) in the healthy family. **(C, D)** Genome coverage of different sequencing depths for CFC518 (C) or CFC2282 (D). **(E, F)** FPR for CFC518 (E) and CFC2282 (F). **(G, H)** ADO of CFC518 (G) or CFC2282 (H). **(I, J)** Genome coverage of the regions corresponding to the 67-gene panel from WGS of CFC518 (I) or CFC2282 (J). **(K, L)** Number of covered 67-gene panel genes in the WGS of CFC518 (K) or CFC2282 (L). **(M, N)** Genome coverage of the whole-exon region in the WGS of CFC518 (M) or CFC2282 (N). **(O, P)** Number of covered whole-exon region genes in the WGS of CFC518 (O) or CFC2282 (P). **(Q, R)** Genome coverage of the OMIM gene panel in the WGS of CFC518 (Q) or CFC2282 (R). **(S, T)** Number of covered OMIM genes in the WGS of CFC518 (S) or CFC2282 (T).

sequencing was performed in addition to 10–15X WGS from the same cell. As a result, we could directly compare the performance of the two methods for investigating the same set of disease-causing genes at the single-cell level (Table 2). In the 67-gene panel region, the genome coverages from the single-cell WGS of the deafness, hemophilia, and LVAS families were 69.3%, 64.8%, and 68.7%, respectively. However, the corresponding coverages using the targeted sequencing method were 60.3%, 56.9%, and 63.1%, respectively.

In the targeted sequencing of the 67 genes, the numbers of genes covered were 62, 59, and 62 in the three disease families. Among them, 58, 59, and 61 genes could be used for the haplotype analysis. By using WGS, 65, 65, and 66 of the 67 genes were covered in the three disease families, and all these genes met the requirements for the haplotype analysis. Overall, these results indicate that the WGS method outperformed targeted sequencing in the 67-gene region.

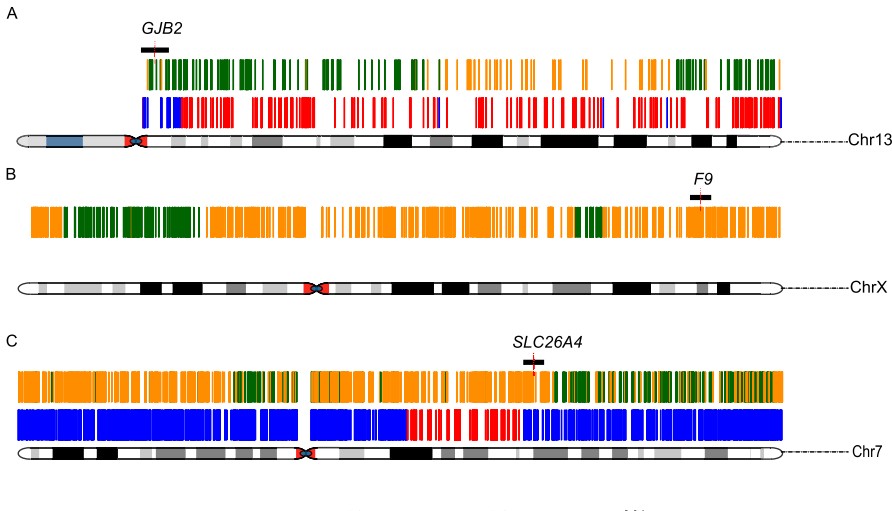

**Figure 2. Haplotype inheritance surrounding disease-causing genes in the three monogenic disease families.**
**(A)** Deafness family with the haplotype P1/M2 in Chr13: *GJB2*. **(B)** Hemophilia family with the haplotype M1 in ChrX:*F9*. **(C)** LVAS family with the haplotypes P1/M1 in Chr7:*SLC26A4*.

With respect to FPR and ADO, we studied CFC178 from the deafness family and CFC616 from the hemophilia family; the LVAS family was excluded because of the lack of an amniotic fluid sample or a fetal villi sample. Overall, the ADO and FPR with WGS (CFC178: 17.3% ADO, 30.22% FPR; CFC616: 18.5% ADO, 26.9% FPR) were better than those with targeted region sequencing (CFC178: 20.6% ADO, 31.7% FPR; CFC616: 16.1% ADO, 30.4% FPR), indicating that the WGS produced more reliable SNP typing.

After haplotype analysis, the number of SNPs ~4 Mb (2-Mb region upstream and downstream) around the pathogenic gene locus identified with WGS (CFC178: 5175; CFC616: 5102; CFC111: 6931) was significantly higher than that identified by panel sequencing (CFC178: 817; CFC616: 682; CFC111: 719). Similarly, more key SNPs were identified upstream and downstream with WGS (CFC178: 16/7; CFC616: 6/7; CFC111: 32/35) than with panel sequencing (CFC178: 4/0; CFC616: 2/0; CFC111: 3/6). More importantly, the key SNPs in the WGS data covered both upstream and downstream of the gene, accurately predicting the haplotype and determining carrier status. However, for two of the three families, targeted sequencing failed to capture any key SNP downstream of the disease-causing genes; thus, the haplotypes of their single cells could not be accurately determined (see the Discussion section). In summary, the WGS method is superior to the targeted panel sequencing in terms of genome coverage of targeted regions, sequencing quality, and haplotype analysis.

### Global genetic risk estimation based on single-cell WGS

We focused on whole human exonic and OMIM regions in the single-cell WGS data to evaluate the possibility of estimating global genetic risk. In brief, the genome coverages in the whole exonic region were 68.2% (CFC178), 60.8% (CFC616), and 70.9% (CFC111) and those in the OMIM gene region were 67.9% (CFC178), 54.6% (CFC616), and 63.2% (CFC111) (Table 2). Furthermore, the number of exome genes covered by WGS was 20,915 (91.9%) for CFC178 from the deafness family, 20,083 (88.2%) for CFC616 from the hemophilia family, and 21,148 (92.9%) for CFC111 from the LVAS family. Among them, the number of genes that

could be used for haplotype analysis was 20,198 (88.7%), 20,014 (87.9%), and 20,945 (92.0%), respectively. Regarding the OMIM genes, 14,071 (92.8%) were covered in the deafness family, 13,342 (87.8%) in the hemophilia family, and 14,319 (94.4%) in the LVAS family. Among them, 13,512 (89.1%), 12,377 (81.6%), and 13,710 (90.3%) could be used for haplotype analysis, respectively. These results indicate that haplotypes for most genes can be predicted by combining single-cell WGS and haplotype analysis.

## Discussion

Prenatal diagnosis is an effective and necessary method for better managing inherited diseases. Invasive prenatal diagnosis methods, for example, amniocentesis, are often risky to pregnant women and fetuses (Agarwal & Alfirevic, 2012; Akolekar et al, 2015). As a supplement or even an alternative to invasive prenatal diagnosis, NIPT is potentially a better candidate for prenatal screening and diagnostics (Lau et al, 2014). Previous NIPT mainly focused on predicting chromosomal diseases (Porreco et al, 2014; Minarik et al, 2015; Zhang et al, 2019). Given the advances in sequencing technologies and data analysis, recent cffDNA-based NIPT studies are shifting the focus toward monogenic diseases (Zhang et al, 2019). However, the performance accuracy of this method is compromised because of the intrinsic limitations of cffDNA (Spits & Sermon, 2009; Beaudet, 2016; Rabinowitz & Shomron, 2020). Thus, new technologies and algorithms are urgently needed to improve the noninvasive prenatal diagnosis of monogenic diseases.

In addition to the use of cffDNA for NIPT, the circulating fetal nucleated red blood cells and cTBs in the maternal blood contain pure and complete fetal genetic information and could also be used for prenatal diagnostics (Choolani et al, 2012; He et al, 2017; Vossaert et al, 2018; Vossaert et al, 2021). The feasibility of cbNIPT in the prenatal detection of monogenic diseases (e.g., cystic fibrosis [Jeppesen et al, 2021] and spinal muscular atrophy [Beroud et al, 2003]) and preimplantation genetic testing have been demonstrated by several groups (Chang et al, 2021; Toft et al, 2021). Because

Table 2.  Sequencing quality assessment of WGS and targeted sequencing in single cells.

| Disease | | Deafness | Hemophilia | LVAS | Health | |
|---|---|---|---|---|---|---|
| Sample | | cTBs-CFC178 | cTBs-CFC616 | cTBs-CFC111 | cTBs-CFC518 | cTBs-CFC2282 |
| Genome coverage (%) | WGS | 56.8 | 51.9 | 57.7 | 53.3 | 54.6 |
| Average depth | WGS | 14.1 | 8.4 | 16.04 | 14.32 | 15.56 |
| | Targeted sequencing | 114 | 88.4 | 110.21 | 103.8 | 96.9 |
| Coverage (%) of the 67-gene panel region | WGS | 69.3 | 64.81 | 68.7 | 67.7 | 67.6 |
| | Targeted sequencing | 60.31 | 56.9 | 63.1 | 59.5 | 62.2 |
| Number of covered 67-gene panel | WGS | 65 | 65 | 66 | 67 | 66 |
| | Targeted sequencing | 62 | 59 | 62 | 62 | 63 |
| Number of available 67-gene panel genes for haplotyping | WGS | 65 | 65 | 66 | | |
| | Targeted sequencing | 58 | 59 | 61 | | |
| Coverage (%) of the whole-exome region | WGS | 68.2 | 60.8 | 70.9 | 66.1 | 63.5 |
| Number of covered exome genes | WGS | 20,915 | 20,083 | 21,148 | 21,378 | 20,747 |
| Number of available exome genes for haplotyping | WGS | 20,198 | 20,014 | 20,945 | | |
| Coverage (%) of the OMIM region | WGS | 67.9 | 54.6 | 63.2 | 65.1 | 65.2 |
| Number of covered OMIM genes | WGS | 14,071 | 13,342 | 14,391 | 14,108 | 14,167 |
| Number of available OMIM genes for haplotyping | WGS | 13,512 | 12,377 | 13,710 | | |
| FPR (%) | WGS | 30.22 | 26.9 | — | 28.9 | 28.6 |
| | Targeted sequencing | 31.7 | 30.4 | — | 32.9 | 28.7 |
| ADO (%) | WGS | 17.3 | 18.5 | — | 17.8 | 19.6 |
| | Targeted sequencing | 20.6 | 16.1 | — | 20.4 | 21.3 |
| SNPs ~4 Mb | WGS | 5175 | 5102 | 6931 | | |
| | Targeted sequencing | 817 | 682 | 719 | | |
| Key SNPs ~4 Mb in the upstream region | WGS | 16 | 6 | 32 | | |
| | Targeted sequencing | 4 | 2 | 3 | | |
| Key SNPs ~4 Mb in the downstream region | WGS | 7 | 7 | 35 | | |
| | Targeted sequencing | 0 | 0 | 6 | | |

Data of the healthy family are represented by results from gradient 4 (15×).

of the lack of interference from restrictive placental mosaicism, the initial experiments focused on the application of circulating fetal nucleated red blood cells (Bianchi et al, 2002). However, probably because of the low abundance or instability of FNRBCs, most studies failed to capture FNRBCs in early pregnancy; thus, follow-up studies were conducted on cTBs and showed that the maternal peripheral blood contains 1–6 cTBs/ml in the first trimester of pregnancy (Oosterwijk et al, 1998; Bianchi et al, 2002). Regarding placental mosaicism, studying more fetal cells with potentially different genotypes from each sample can promote the accuracy of monogenic disease diagnostics using cbNIPT (Breman et al, 2016;

Vossaert et al, 2019). In the future, it will be necessary to increase the number of cTBs by improving the efficiency of cell capture to ensure accurate prenatal diagnosis. In addition, most previous cbNIPT studies only focused on chromosomal (Breman et al, 2016) and subchromosomal (Kolvraa et al, 2016) abnormalities, and it remains largely unknown whether cbNIPT can be widely used for the prenatal diagnosis of monogenic diseases.

In our previous study (Chang et al, 2021), we used a combination of known markers, such as cytokeratin as a positive marker and CD45 as a negative marker (Hatt et al, 2014), to label cTBs as target fetal cells. In addition, single-cell STR analysis confirmed the source

of isolated cTBs, indicating the feasibility of isolating cTBs from maternal blood. To avoid potential amplification errors and ADO during WGA (Liu et al, 2018), we subsequently took advantage of targeted sequencing of a 67-gene panel combined with haplotype analysis to detect monogenic diseases. In all deafness and ichthyosis disease cases, we successfully determined the inherited haplotypes of the fetus. However, some genes in the panel cannot be parsed because of ADO. In addition, targeted panel sequencing requires prior knowledge of disease-causing genes and mutations, limiting its application for the global estimation of genetic risk.

In the current study, we further explored the possibility of combining WGS with haplotype analysis for better prenatal diagnostics. Our results again demonstrated that cTBs isolated from maternal peripheral blood are sufficient for downstream genetic analyses to diagnose monogenic diseases. In addition, WGS combined with haplotype analysis successfully determined the genotype of the pathogenic gene in a fetus based on captured CFCs. Comparing WGS and targeted panel sequencing showed that WGS is superior to the targeted sequencing approach we previously used in terms of genome coverage, the number of SNP sites covering ~4 Mb, the ADO ratio, and the FPR. Most importantly, more key SNPs were present in the WGS data than in the targeted sequencing, and WGS covered regions both upstream and downstream of the gene. These data highlight the value of WGS for precise haplotyping and prenatal diagnostics.

Furthermore, we focused on exome and OMIM regions to investigate the feasibility of genome-wide genetic risk evaluation. In general, single-cell WGS data showed 60–70% genome coverage for the exome and OMIM regions. However, haplotype analysis could be performed for ~90% of the exome or OMIM genes (the upstream and downstream regions contained at least one key SNP), highlighting the possibility of diagnosing most genetic risks by combining single-fetal WGS data with familial haplotypes. Similar to cffDNA-based NIPT, fetoplacental mosaicism could be a major confounding factor for accurate diagnosis (Mardy & Wapner, 2016; Hartwig et al, 2017; Van Opstal et al, 2020; Rosner et al, 2021; Vossaert et al, 2021). In addition, the current method requires the proband for the haplotyping analysis, which limits its application for some affected families. More efficient fetal cell capture, better analysis algorithms, and more clinic data are required before the application of cbNIPT in prenatal testing or diagnostics. To the best of our knowledge, this is the first study to investigate the application of WGS from single captured cTBs in the prenatal diagnosis of monogenic disease and to estimate genetic risk on a whole-genome level. Overall, our study provides a novel and feasible NGS-based cbNIPT solution for targeted and global estimations of prenatal health. In the future, more clinical studies are required to investigate the feasibility of this method in diagnosing various genetic abnormalities and comprehensively evaluating fetal health.

# Materials and Methods

### Patient recruitment

The patient recruitment criteria were as follows: age between 20 and 45 yr; body mass index in the range of 18–25 kg/m$^2$; 11–16 wk pregnant;

family or one of the parents having a disease-causing genetic mutation, and the proband with the disease-causing mutation has been characterized (preferred). Women with one or more of the following conditions were excluded: fetus died in the uterus before sampling, no signed informed consent documentation, incomplete sampling, requested to withdraw from the study, and other circumstances that may affect the test results. A pregnant woman from a healthy family was also recruited as a control. This study was approved by the Scientific Research Ethical Committee of Peking University Third Hospital (approval reference number 2019-246-02). All subjects participating in the project signed an informed consent form.

### Samples collection

Maternal peripheral blood (15 ml) was drawn during pregnancy at 11–16 wk with an LBgard Blood Tube (Biomatrica) for fetal trophoblast cell capture and gDNA extraction. Amniotic fluid was collected for fetal chorionic cell capture and validation if applicable. Buccal swabs were taken at least three months after birth for gDNA extraction. Peripheral blood (4 ml in an EDTA anticoagulant tube) from the spouse and/or proband (if applicable) was collected for gDNA extraction.

### cTB enrichment and isolation

cTB enrichment and isolation were conducted, as previously described (Chang et al, 2021). Briefly, nucleated cells were collected by centrifugation with lymphocyte separation medium (density 1.077) after maternal blood collection. Subsequently, all nucleated cells were enriched using a cocktail of bead-linked antibodies (CD105, CD141, and HLA-G). The enriched cells were then stained with fluorescently labeled antibodies including DAPI (422801; BioLegend), anti-cytokeratin FITC (clone C11 628608; BioLegend), and anti-CD45 PE (clone 2D1 368510; BioLegend). Finally, we analyzed and screened candidate trophoblast single cells that met the selection criteria (DAPI positive, keratin positive, CD45 negative) using the UniPicker instrument (Unimed Biotech) for the subsequent steps. The sample processing is illustrated in Fig S3.

### Whole-genome DNA amplification of single cells and QC process

A PicoPLEX WGA Kit (R300672; Takara Bio) was used to amplify the whole genome of captured single cells. The WGA products were purified with DNA Clean Beads (N411-02; VAHTS) and stored at −20°C for further analysis. ~1.5 µg of DNA products (~200–1,000 bp in length) were recovered after WGA. gDNA was extracted with 200 µl of blood from the parents using a QIAGEN DSP Blood Mini Kit (61104; QIAGEN) according to the manufacturer's directions. A genotyping assay was performed using an AmpFlSTR Identifiler Plus Kit (4427368; Applied Biosystems) containing 13 of the required loci from the Combined DNA Index System and three additional loci (D2S1338, D19S433, and the amelogenin gender-determining marker) to confirm the fetal origin of the isolated single cells. Briefly, STR–PCR was performed with the WGA products and the gDNAs from the parents. The PCR products were then subjected to capillary electrophoresis using an Applied Biosystems 3130xl Genetic Analyzer, and the genotypes were analyzed by GeneMapper 3.2 analysis software.

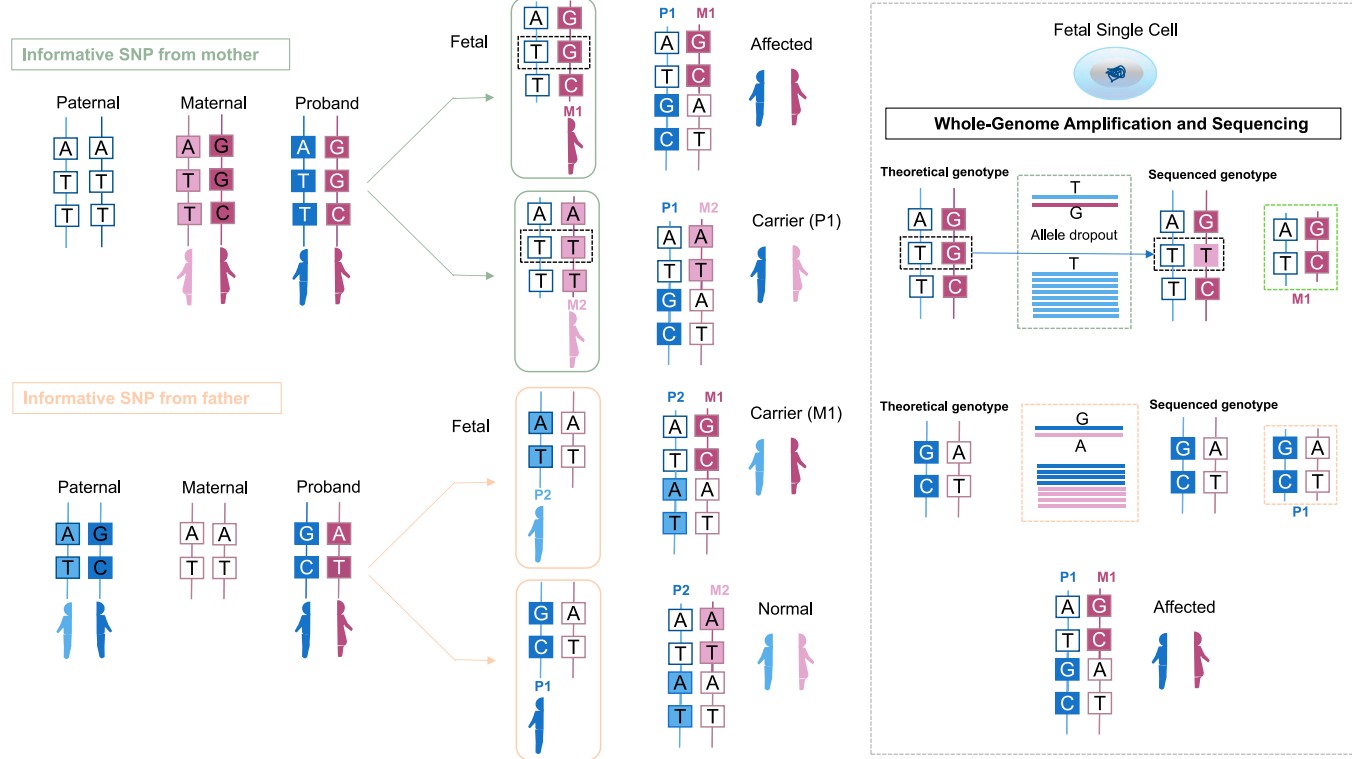

**Figure 3.  Workflow of embryonic haplotype construction using the genotypes of the father, mother, and proband.**
The details are described in the Materials and Methods section.

## Single-cell targeted sequencing

Panel design and library preparation details are described in our previous study (Chang et al, 2021). Briefly, 2 µg DNA libraries were hybridized to the custom panel according to the manufacturer's instructions (Twist). Captured DNA libraries were sequenced using 150-bp paired-end index sequencing on a Hiseq 2500 (Illumina) instrument per the manufacturer's instructions.

## Alignment, SNP/indel calling, and sequencing quality assessment

Adapters were trimmed from raw reads using fastp (Chen et al, 2018), and the reads were aligned to the reference genome (Hg19) using Burrows–Wheeler alignment (Li & Durbin, 2009) with the default parameters. PCR duplications were removed by Picard (https://broadinstitute.github.io/picard/). After filtering low-quality reads with mapping quality <10 and base quality <15, we recalibrated the base qualities and performed the SNP calling using the Genome Analysis Toolkit (GATK; filtering criteria: coverage >10 and quality value >20) (McKenna et al, 2010). We used the sequencing results from the father, mother, and proband to construct informative SNPs and marked the loci that may be affected by ADO as non-key SNPs; the others were designated key SNPs. The ADO ratio and FPR were calculated for the single-cell sequencing data from WGA products. The ADO ratio was calculated by the following equation:

$$\text{ADO ratio} = \frac{\sum \text{SNP}_d}{\text{Total SNPs}},$$

where SNP is the allele for which the amniotic fluid sample is heterozygous, whereas the single-cell sample is homozygous.

The FP ratio was calculated by the following equation:

$$\text{False Positive ratio} = \frac{\text{FP (false positive)}}{\text{Total SNPs}},$$

where FP indicates the number of loci with different genotypes between the control and single-cell samples.

Panel region coverage was calculated using single-cell WGS data by the following equation:

$$\text{Panel coverage} = \frac{\sum \text{covered region}}{\text{Total panel region}}.$$

This formula applies to the 67-gene region and the WES and OMIM gene regions.

In addition, analyses of the genome coverage by WGS in the gene region, the number of covered genes (≥1 reads), and the number of genes that could be used for haplotype analysis (both upstream and downstream contain at least one key SNP) were also performed in the WES (Agilent v6) and OMIM regions (https://www.omim.org/, version 2020.05.25, with a total of 15,145 genes).

## Library construction and WGS

Single-cell WGA products and gDNAs were sheared with Covaris to obtain fragments ranging from 250–350 bp in size. Paired-end sequencing libraries were prepared with KAPA Hyper Prep Kit (KK8504; Kapa Biosystems) as described in the manufacturer's protocol. Barcodes were introduced during index adapter ligation for multiplex sequencing. DNA libraries were measured with an Agilent 2100 bioanalyzer (Agilent) for insert size and quantified by Qubit.

DNA libraries were sequenced using 150-bp paired-end index sequencing on a Hiseq 2500 (Illumina) device according to the manufacturer's instructions. For WGS data from single-cell WGA products and parental gDNA from the healthy families, a total of eight gradients of randomly subsampled reads at 1X, 5X, 10X, 15X, 20X, 25X, 30X, and 35X human genome coverage were used to estimate the effect of sequencing depth on further analysis. The appropriate sequencing depth was selected based on the depth, FPR, ADO, and coverage of the 67-gene panel region; coverage of the whole-exon region; and coverage of the OMIM gene region for WGS on captured fetal trophoblast cells from healthy families.

## Single-cell haplarithmisis

We combined the sequencing results of fetal trophoblast cells and the corresponding father, mother, and proband into a family for haplotype analysis (Handyside et al, 2010) (Fig 3). Assuming there is one homologous recombination on each chromosome inherited by the proband, the locus where the paternal genotype is AB and the maternal genotype is non-AB is IFF (informative SNP from father) and the locus where the maternal genotype is AB and the paternal genotype is non-AB is IFM (informative SNP from mother). We used the sequencing results of the father, mother, and proband to construct informative SNPs and marked the loci that may be affected by ADO as non-key SNPs, whereas the others were designated key SNPs. If the proband carried the haplotype of the paternal causative gene, it was marked as P1; otherwise, it was marked as P2. Similarly, if the proband carried the haplotype of the maternal causative gene, it was marked M1; otherwise, it was marked M2. Therefore, there are four possible combinations of haplotypes for the fetal trophoblast cells: P1/M1 (affected), P1/M2 (carrier from father), P2/M1 (carrier from mother), and P2/M2 (normal). Haplotype inheritance figures were drawn with all informative SNPs by matplotlib (https://github.com/matplotlib/matplotlib) according to the single-cell haplotype analysis results.

# Data Availability

The genome sequencing data for this publication have been deposited to the NCBI BioProject database (http://www.ncbi.nlm.nih.gov/bioproject/883017) under accession number PRJNA883017.

# Supplementary Information

# Acknowledgements

This study was approved by the Scientific Research Ethical Committee of Peking University Third Hospital (approval reference number 2020-191-02). We would like to thank all of the subjects who participated in the program and signed an informed consent form. This work was sponsored by the Natural Science Foundation of Beijing Municipality (No. 7202226) and funding from National Clinical Research Center for Obstetrics and Gynecology (Peking University Third Hospital) (No. BYSYSZKF2021004).

## Author Contributions

L Chang: conceptualization, resources, data curation, formal analysis, funding acquisition, validation, investigation, visualization, methodology, and writing—original draft, review, and editing.
H Jiao: conceptualization, resources, data curation, formal analysis, validation, investigation, visualization, methodology, and writing—original draft, review, and editing.
J Chen: data curation, software, formal analysis, methodology, and writing—original draft, review, and editing.
G Wu: formal analysis, investigation, visualization, and writing—original draft, review, and editing.
P Liu: resources, investigation, methodology, and writing—review and editing.
R Li: resources, investigation, methodology, and writing—original draft, review, and editing.
J Guo: resources, investigation, methodology, and writing—review and editing.
W Long: validation, methodology, and writing—review and editing.
X Tang: validation, methodology, and writing—review and editing.
B Lu: formal analysis, visualization, and writing—review and editing.
H Xu: validation, methodology, and writing—review and editing.
H Wu: conceptualization, resources, data curation, formal analysis, validation, investigation, visualization, methodology, project administration, and writing—original draft, review, and editing.

## Conflict of Interest Statement

The authors declare that they have no conflict of interest.

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
