## [Reviewer comments · Life Science Alliance]

Life Science Alliance

Single-cell whole-genome sequencing, haplotype analysis in prenatal diagnosis of monogenic diseases

Liang Chang, Haining Jiao, Jiucheng Chen, Guanlin Wu, Ping Liu, Rong Li, Jianying Guo, Wenqing Long, Xiaojian Tang, Bingjie Lu, Haibin Xu, and Han Wu

DOI: <https://doi.org/10.26508/lsa.202201761>

Corresponding author(s): Liang Chang, Peking University Third Hospital and Han Wu, Lawrence Berkeley National Laboratory

Review Timeline:

Submission Date:	2022-10-08
Editorial Decision:	2022-12-03
Revision Received:	2023-01-18
Editorial Decision:	2023-02-03
Revision Received:	2023-02-10
Accepted:	2023-02-10

Scientific Editor: Novella Guidi

Transaction Report:

December 3, 2022

Re: Life Science Alliance manuscript #LSA-2022-01761-T

Chang Liang
Peking University Third Hospital

Dear Dr. Liang,

Thank you for submitting your manuscript entitled "Single cell whole-genome sequencing and haplotype analysis in prenatal diagnosis of monogenic diseases" to Life Science Alliance. The manuscript was assessed by expert reviewers, whose comments are appended to this letter. We invite you to submit a revised manuscript addressing the Reviewer comments.

Thank you for this interesting contribution to Life Science Alliance. We are looking forward to receiving your revised manuscript.

Sincerely,

B. MANUSCRIPT ORGANIZATION AND FORMATTING:

Reviewer #1 (Comments to the Authors (Required)):

Non-invasive prenatal screening has changed the nature of prenatal genetic testing over the last 10 years. Whereas the main NIPS approach is focusing on cfDNA, a niche of teams is zooming in on the circulating fetal cells in the maternal blood. The main advantage of using cells is that a whole genome analysis would become possible. In this article, the authors explore the use of whole genome amplification and sequencing as an approach to deduce the haplotypes of the fetus. Whereas previously different markers have shown the presence of the fetus, the correct mutational analysis remains a huge challenge. The proof-of-concept presented here might alleviate those difficulties and could push the field forward. This article and the methodological description advances the field. The article is clear and well written.

Minor comments:

The approaches used and explored in the manuscript build on the pioneering work of the preimplantation genetic diagnosis field. The references to some of those original papers are lacking. E.g. haplarithmisis is used, but without referencing the original papers.

Table 1 is sloppy (but this might be purely an editing issue)

Reviewer #2 (Comments to the Authors (Required)):

In the paper called "Single cell whole-genome sequencing and haplotype analysis in prenatal diagnosis of monogenic diseases", authors describe their experience in developing a non-invasive prenatal analysis for monogenic disorders. The test is based on single-cell WGS and STR-based haplotyping of circulating placenta-derived cells in the maternal plasma. In this study, they included three affected families (deafness, LVAS and hemophilia) and one control - healthy family. They also compare targeted panel sequencing of 67 genes and WGS on single cells.

This study provides interesting insights on feasibility of circulating cell-based analysis, which in recent years has gained popularity in prenatal diagnostics research. However, the lack of scientific writing style and consistency (e.g., differences in naming of included families, cells) makes the manuscript and its scientific content very difficult to follow or appreciate. The professional editing or scientific writing services would be highly recommended and would undoubtedly increase the value of the paper, which would help in publishing it.

The conclusion in the abstract stating the possibility to detect "most monogenic diseases" does not fit with the scope of the paper, as it barely analyses three different monogenic diseases. A larger study with analysis of many other monogenic disorders is needed to support those claims.

Additionally, trophoblasts are not fetal cells (as written in the manuscript) and they pose the same risks of mosaic as cffDNA or CVS, and that is why NIPT is not a diagnostic tool yet, which should also be addressed in the manuscript.

After the necessary changes are made, the study would be a good addition to the new and fast-growing research field of non-invasive prenatal analysis.

We would like to thank the reviewers for their positive feedback and their thoughtful comments for improvement of the manuscript. We have now revised the manuscript, and carefully addressed every comments and suggestions raised by the referees. Given the nature of the reviewers' comments, most of these changes were minor adjustments to the text or figures. All revisions were marked in red in the new manuscript.

Reviewer 1

Reviewer #1, Overview: Non-invasive prenatal screening has changed the nature of prenatal genetic testing over the last 10 years. Whereas the main NIPS approach is focusing on cfDNA, a niche of teams is zooming in on the circulating fetal cells in the maternal blood. The main advantage of using cells is that a whole genome analysis would become possible. In this article, the authors explore the use of whole genome amplification and sequencing as an approach to deduce the haplotypes of the fetus. Whereas previously different markers have shown the presence of the fetus, the correct mutational analysis remains a huge challenge. The proof-of-concept presented here might alleviate those difficulties and could push the field forward. This article and the methodological description advances the field. The article is clear and well written.

RESPONSE: We would like to thank the reviewer for positive comments and highlighting the significance of this study. That's a great encouragement for our further research to improve our methodology for rare cell capture and to expand our clinic study in monogenic disorders.

Reviewer #1, Minor comments 1: The approaches used and explored in the manuscript build on the pioneering work of the preimplantation genetic diagnosis field. The references to some of those original papers are lacking. E.g. Haplarithmisis is used, but without referencing the original papers.

RESPONSE: Thank you for pointing out the lacking of references, the following related references have been added to the manuscript and marked in red both in the body and the reference sections.

Line74-75 (Steele MW & Breg WR, Jr., 1966, Nadler HL & Gerbie AB, 1970, Simoni G et al, 1983, Levy B & Stosic M, 2019)

Line77-78 (Simpson JL, 2012, Alfirovic Z et al, 2017, Salomon LJ et al, 2019)

Line93-94 (Zipursky A et al, 1959, Walknowska J et al, 1969, Herzenberg LA et al, 1979, Beaudet AL, 2016, Singh R et al, 2017)

Line99 (Kolvræa S et al, 2016, Hou S et al, 2017, Vossaert L et al, 2019)

Line112 (Zhang CZ et al, 2015, Volozonoka L et al, 2022)

Line115 (Clark AG, 1990)

Line260-261 (Lau TK et al, 2014)

Line262 (Porreco RP et al, 2014, Minarik G et al, 2015, Zhang J et al, 2019)

Line324-325 (Mardy A & Wapner RJ, 2016, Hartwig TS et al, 2017, Van Opstal D et al, 2020, Rosner M et al, 2021, Vossaert L et al, 2021)

Line435-436 (Handyside AH et al, 2010)

Reviewer #1, Minor comments 2: Table 1 is sloppy (but this might be purely an editing issue).

RESPONSE: Sorry for the sloppy Table 1, we have adjusted it, see Table 1.

Reviewer 2

Reviewer #2, Overview: In the paper called "Single cell whole-genome sequencing and haplotype analysis in prenatal diagnosis of monogenic diseases", authors describe their experience in developing a non-invasive prenatal analysis for monogenic disorders. The test is based on single-cell WGS and STR-based haplotyping of circulating placenta-derived cells in the maternal plasma. In this study, they included three affected families (deafness, LVAS and hemophilia) and one control - healthy family. They also compare targeted panel sequencing of 67 genes and WGS on single cells.

RESPONSE: Thank you for the positive feedback and helpful comments on our study.

Reviewer #2, point 1: This study provides interesting insights on feasibility of circulating cell-based analysis, which in recent years has gained popularity in prenatal diagnostics research. However, the lack of scientific writing style and consistency (e.g., differences in naming of included families, cells) makes the manuscript and its scientific content very difficult to follow or appreciate. The professional editing or scientific writing services would be highly recommended and would undoubtedly increase the value of the paper, which would help in publishing it.

RESPONSE: Thank you for your comment. We are sorry for bad English expression. We have tried our best to make necessary changes and sent the manuscript to an editing company (letpub) for English language editing during the revision process. All of the families and cells mentioned in the manuscript have been described in a consistent order and names. The corresponding figures and tables have also been modified. Please refer to the following locations in the revised manuscript for more details:

1.Line 50-56(abstract)

2.Line 128-129, 139-164, 206-208 (results)

3.Fig 1, S2 (Figure)

4.Table 1, S1-S3, S5-S7 (Tables)

5.Line 697-736, Fig 1-3, S1-S3 (Figure legends)

Reviewer #2, point 2: The conclusion in the abstract stating the possibility to detect "most monogenic diseases" does not fit with the scope of the paper, as it barely analyses three different monogenic diseases. A larger study with analysis of many other monogenic disorders is needed to support those claims.

RESPONSE: Thank you for your comment. We agree with the reviewer of comment that there is indeed insufficient evidence to draw a conclusion, and we have made changes in the abstract at **line 59 - 60.**

Reviewer #2, point 3 Additionally, trophoblasts are not fetal cells (as written in the manuscript) and they pose the same risks of mosaic as cffDNA or CVS, and that is why NIPT is not a diagnostic tool yet, which should also be addressed in the manuscript.

After the necessary changes are made, the study would be a good addition to the new and fast-growing research field of non-invasive prenatal analysis.

RESPONSE: The reviewer made a good point on trophoblast cells as we briefly mention in the old version. We have further emphasized the potential mosaic problem with trophoblast cells and the limitations of trophoblast cell-based prenatal testing method. Please refer to the following modifications in the revised manuscript for more clarification:

1.Line 52-54 (abstract)

2.Line 90-94 (introduction)

3.Line 141-165, 170-171 (results)

4.Line 270, 294-296, 323-327 (discussion)

5.Line 350, 356, 431-432, 434 446(materials and methods)

6.Line 703-736 (Figure legends)

7.Table1, 2 (tables)

February 3, 2023

RE: Life Science Alliance Manuscript #LSA-2022-01761-TR

Prof. Chang Liang
Peking University Third Hospital
49 North Garden Rd., Haidian District
Beijing, Beijing 100191
China

Dear Dr. Liang,

Thank you for submitting your revised manuscript entitled "Single-cell whole-genome sequencing, haplotype analysis in prenatal diagnosis of monogenic diseases". We would be happy to publish your paper in Life Science Alliance pending final revisions necessary to meet our formatting guidelines.

- please address the remaining Reviewer'2 comments
- please add ORCID ID for secondary corresponding author-they should have received instructions on how to do so
- please add the Twitter handle of your host institute/organization as well as your own or/and one of the authors in our system

Figure Check:

- please add scale bars to Figure S2A and to the top right image in Figure S3

A. FINAL FILES:

B. MANUSCRIPT ORGANIZATION AND FORMATTING:

Sincerely,

Reviewer #2 (Comments to the Authors (Required)):

The manuscript "Single cell whole-genome sequencing, haplotype analysis in prenatal diagnosis of monogenic diseases" presents the authors' development of a non-invasive prenatal diagnosis approach for inherited monogenic disorders. The test utilizes single-cell WGS of placental-derived cells found in maternal plasma and haplotyping to detect whether the pathogenic variant is inherited. The study included three affected families, as well as a healthy control family. The authors also compare the results of targeted panel sequencing of 67 genes with those of WGS on circulating trophoblast cells (CTCs), reporting the benefit of employing WGS. This study offers intriguing findings on the usability of CTC from maternal plasma analysis in prenatal diagnostics, which shows a great potential for detection of inherited monogenic disorders.

Minor comments:

1. Needs be noted in the manuscript that the method is suitable for inherited variant detection, as well as possible diagnoses of inherited monogenic disorders from the parents.
2. In the abstract, the circulating cells used for the study are described as circulating trophoblast cells (cTBs), which is a correct term. However, later in the main text they are described as either fetal trophoblast cells, or circulating trophoblasts, or fetal cell, finally coming back to cTBs in the discussion. This inconsistency in proper naming might cause confusion about which fetus-representing cells were actually used in this study.
3. Thorough inspection of abbreviations used in the manuscript is recommended. As a standard rule of scientific writing, all abbreviations should be written out in full on the first use, both in the abstract and in the main text itself. For examples, in lines 94, 134, 168,
4. For the limitations of the method, it should be mentioned that another proband (child) is needed for the haplotyping analysis, making it not suitable for a portion of the families.
5. Figure legends are provided in the erratic way, starting with supplementary figures. Bellow, the provided figures did not have any indication of which figure it was and which legend belong to it. I understand that it could be the fault of the reviewing portal itself, however, it is something to look into for the final manuscript.
6. Additionally, Table 1 is still difficult to read - also, probably the portal/editing issue, but should be improved.

Overview

We would like to thank the reviewers for their positive feedback and thoughtful comments on how to improve the manuscript. We have now revised the manuscript and carefully addressed all the comments and suggestions made by the reviewers. Given the nature of the reviewers' comments, most of these changes were minor adjustments to the text or figures. All revisions have been marked in red in the new manuscript.

Reviewer 2

Reviewer #1, Overview: The manuscript "Single cell whole-genome sequencing, haplotype analysis in prenatal diagnosis of monogenic diseases" presents the authors' development of a non-invasive prenatal diagnosis approach for inherited monogenic disorders. The test utilizes single-cell WGS of placental-derived cells found in maternal plasma and haplotyping to detect whether the pathogenic variant is inherited. The study included three affected families, as well as a healthy control family. The authors also compare the results of targeted panel sequencing of 67 genes with those of WGS on circulating trophoblast cells (CTCs), reporting the benefit of employing WGS. This study offers intriguing findings on the usability of CTC from maternal plasma analysis in prenatal diagnostics, which shows a great potential for detection of inherited monogenic disorders.

RESPONSE: We would like to thank the reviewer for positive comments and highlighting the significance of this study. That's a great encouragement for our further research to improve our methodology for rare cell capture and to expand our clinic study in monogenic disorders.

Reviewer #1, Minor comments 1: Needs be noted in the manuscript that the method is suitable for inherited variant detection, as well as possible diagnoses of inherited monogenic disorders from the parents.

RESPONSE: Thank you for your comment. We agree with the reviewer that this method can potentially diagnose monogenic disorders inherited from the parents, and we have already stated that in several paragraphs at **line 312-314, line324-329 and line 335-342.**

Reviewer #1, Minor comments 2: In the abstract, the circulating cells used for the study are described as circulating trophoblast cells (cTBs), which is a correct term. However, later in the main text they are described as either fetal trophoblast cells, or circulating trophoblasts, or fetal cell, finally coming back to cTBs in the discussion. This inconsistency in proper naming might

cause confusion about which fetus-representing cells were actually used in this study.

RESPONSE: Thank you for your comment. All circulating cells mentioned in the manuscript have been modified accordingly. Please refer to the following locations in the revised manuscript for more details:

1. **Line 94,106,124(introduction)**
2. **Line 141,145,155,156,158,166,174(result)**
3. **Line 277,288,302,313,337(discussion)**
4. **Line 364,365(materials and methods)**
5. **Line 735,740, 741,744,749,751(figure legends)**
6. **Table 1,2**

Reviewer #1, Minor comments 3: Thorough inspection of abbreviations used in the manuscript is recommended. As a standard rule of scientific writing, all abbreviations should be written out in full on the first use, both in the abstract and in the main text itself. For examples, in lines 94, 134, 168.

RESPONSE: Thank you for your comment. We have checked all the abbreviations in the manuscript and made changes accordingly.

1. **Line 94**
2. **Line 106**
3. **Line 133**
4. **Line 165**
5. **Line 177**

Reviewer #1, Minor comments 4: For the limitations of the method, it should be mentioned that another proband (child) is needed for the haplotyping analysis, making it not suitable for a portion of the families.

RESPONSE: Thank you for your comment. We have changed the expression on **line 332-333**.

Reviewer #1, Minor comments 5: Figure legends are provided in the erratic way, starting with supplementary figures. Bellow, the provided figures did not have any indication of which figure it was and which legend belong to it. I understand that it could be the fault of the reviewing portal itself, however, it is something to look into for the final manuscript.

RESPONSE: Thank you for your comment. We have undated the order of the Figure legends.

Reviewer #1, Minor comments 6: Additionally, Table 1 is still difficult to read - also, probably the portal/editing issue, but should be improved.

RESPONSE: Thank you for your comment. We have reformatted all the tables to make them easier to read.

February 10, 2023

RE: Life Science Alliance Manuscript #LSA-2022-01761-TRR

Prof. Liang Chang
Peking University Third Hospital
49 North Garden Rd., Haidian District
Beijing, Beijing 100191
China

Dear Dr. Chang,

Thank you for submitting your Research Article entitled "Single-cell whole-genome sequencing, haplotype analysis in prenatal diagnosis of monogenic diseases". It is a pleasure to let you know that your manuscript is now accepted for publication in Life Science Alliance. Congratulations on this interesting work.

DISTRIBUTION OF MATERIALS:

Again, congratulations on a very nice paper. I hope you found the review process to be constructive and are pleased with how the manuscript was handled editorially. We look forward to future exciting submissions from your lab.

Sincerely,
